# Traffic accident risk prediction based on deep learning and spatiotemporal features of vehicle trajectories

Hao Li[1,2], Linbing Chen [1,2]*

**1** School of Civil Engineering Architecture and the Environment, Hubei University of Technology, Wuhan, China, **2** Key Laboratory of intelligent Health Perception and ecological restoration of rivers and Lakes, Ministry of education, Hubei University of Technology, Wuhan, China

* 648838117@qq.com

## Abstract

With the acceleration of urbanization and the increase in traffic volume, frequent traffic accidents have significantly impacted public safety and socio-economic conditions. Traditional methods for predicting traffic accidents often overlook spatiotemporal features and the complexity of traffic networks, leading to insufficient prediction accuracy in complex traffic environments. To address this, this paper proposes a deep learning model that combines Convolutional Neural Networks (CNN), Long Short-Term Memory networks (LSTM), and Graph Neural Networks (GNN) for traffic accident risk prediction using vehicle spatiotemporal trajectory data. The model extracts spatial features such as vehicle speed, acceleration, and lane-changing distance through CNN, captures temporal dependencies in trajectories using LSTM, and effectively models the complex spatial structure of traffic networks with GNN, thereby improving prediction accuracy. The main contributions of this paper are as follows: First, an innovative combined model is proposed, which comprehensively considers spatiotemporal features and road network relationships, significantly improving prediction accuracy. Second, the model's strong generalization ability across multiple traffic scenarios is validated, enhancing the accuracy of traditional prediction methods. Finally, a new technical approach is provided, offering theoretical support for the implementation of real-time traffic accident warning systems. Experimental results demonstrate that the model can effectively predict accident risks in various complex traffic scenarios, providing robust support for intelligent traffic management and public safety.

## Introduction

With the growth in vehicle ownership and the rapid development of the road transportation industry, the economy has been significantly boosted and is gradually prospering. However, this trend has also led to increasingly severe challenges in road traffic safety.

**Data availability statement:** All relevant data are within the manuscript.

**Funding:** The author(s) received no specific funding for this work.

**Competing interests:** The authors have declared that no competing interests exist.

In 2023, approximately 10 million traffic accidents occurred across China. Among these, there were 436 major road traffic accidents (involving three or more fatalities per incident). Throughout the year, traffic accidents resulted in about 100,000 deaths and approximately 2 million injuries, with direct economic losses reaching around 10 billion RMB [1]. As of 2023, among the top five countries in terms of vehicle ownership, China's traffic accident mortality rate (0.0188%) is significantly higher than that of the United States (0.0124%), Germany (0.0042%), Japan (0.0038%), and India (0.0173%) [2]. Traffic safety has become a major issue affecting China's socio-economic development [3]. Therefore, conducting research on road traffic accident risk prediction is of great significance for building an effective traffic safety warning system. This not only helps identify and eliminate potential safety hazards but also effectively prevents and reduces the occurrence of traffic accidents, thereby ensuring public safety. By predicting high-risk factors for traffic accidents in advance, it provides a scientific basis for formulating corresponding safety measures and management strategies, effectively improving road traffic safety and smoothness.Traffic accident risk prediction relies on observational data from past traffic accidents, aiming to predict the likelihood of accidents occurring at specific locations in the future. In addition to historical accident data, vehicle trajectory data has also become an important source of information, as it contains a wealth of driving behavior patterns and environmental factors. These data provide new perspectives and methods for predicting traffic accident risks, enabling more accurate assessments of potential danger zones and helping to implement more effective preventive measures [4].

Data mining technology based on vehicle trajectories has shown broad application prospects in fields such as traffic management and assisted driving [5]. Vehicle trajectory data contains rich information about driving behavior and environmental conditions. After analysis, it can effectively predict accident risks in complex traffic scenarios. However, traditional traffic management systems often rely on questionnaires. These are mainly used to collect information on travel demand and road traffic conditions. Although these methods are useful for academic research, they are time-consuming, labor-intensive, and limited in data volume. They also lack timeliness and accuracy. The development of sensor technology, wireless communication, and the Internet of Things (IoT) has provided a basis for collecting large-scale data through sensors [6]. This study primarily collects vehicle trajectory data using professional equipment and remote video surveillance.

Researchers have been dedicated to improving the accuracy and effectiveness of traffic accident risk prediction. Wang and Shi developed a short-term speed prediction model using Support Vector Machines (SVM) and hybrid wavelet analysis. Although this model could validate its accuracy to some extent, it has significant limitations when handling spatiotemporal dependencies and dynamic environments [7]. Qing employed a fuzzy mean algorithm and microwave detectors to collect vehicle-related data. Using time segmentation, combined with a Random Forest algorithm, they established an efficient prediction model. While the model performs well in classification accuracy and runtime, it fails to effectively capture the global spatial dependencies, resulting in lower accuracy for traffic accident risk prediction [8].

Wenhua optimized the K-means clustering algorithm and enhanced robustness through fuzzy clustering, and used Particle Swarm Optimization (PSO) to optimize SVM for traffic risk prediction. Although the combination of algorithms improved prediction accuracy, it still fell short in deep fusion of spatiotemporal features and global dependency modeling, especially in dynamic traffic flow predictions, where accuracy was lower than deep learning models [9]. Senyin used wide-area radar detectors to collect vehicle data and classified it based on geographical location. However, in the face of dynamic traffic flow and complex environments, the spatiotemporal classification accuracy was low, and it failed to effectively process global spatiotemporal dependencies [10].

Pan and colleagues proposed a Particle Swarm Optimization Long Short-Term Memory (PSO-LSTM) model based on V2X traffic flow information to predict future vehicle speeds. While this model handled time-series data effectively, its treatment of spatial features remained limited and failed to capture global spatiotemporal dependencies well [11]. Li and others proposed the CDBL framework, which maintains good prediction performance over extended periods. However, it lacks depth and breadth in handling the spatiotemporal features of traffic risks, making it difficult to cope with complex dynamic environments [12].

Ali and colleagues proposed the Dynamic Deep Hybrid Spatiotemporal Neural Network (DHSTNet) model, which efficiently predicted traffic flow in urban areas. However, it still has shortcomings in global spatial dependency modeling and deep fusion of spatiotemporal features [13–15]. Li introduced a Spatiotemporal Graph Convolutional Network (STEGCN) that integrates location and time features, using graph convolution and 1D convolution modules to improve traffic prediction accuracy. However, its adaptability and accuracy improvement in complex traffic scenarios remain limited [16].

Liu and others proposed a Robust Spatiotemporal Graph Convolutional Network (RT-GCN) that enhances robustness through an attention mechanism. While it has advantages in noise handling and missing data, its processing of global spatiotemporal features remains constrained [17]. Fan and colleagues introduced a Random Graph Diffusion Attention Network (RGDAN), which effectively captured spatiotemporal correlations, but the model's accuracy improvement was limited, increasing by only 2%-5% compared to existing deep learning methods [18].

Wang and colleagues proposed the STTN, which combines a Spatiotemporal Transformer Network to improve the accuracy of conflict risk propagation. However, its adaptability in complex traffic network topologies and spatial dependency handling is still limited. GNN performs better in capturing spatial relationships and node dependencies in traffic networks [19]. AL-BAHRI et al. researched the interaction of augmented reality and drone control, optimizing communication network parameters. However, they did not sufficiently consider spatiotemporal feature fusion in complex traffic scenarios [20].

Alblushi and others reviewed the security and system hierarchy of IoT but did not delve into spatiotemporal data processing for traffic accident risk prediction [21]. Zakarya and colleagues proposed a service migration strategy that optimized edge computing, but they did not address spatiotemporal data and multi-level feature fusion in traffic accident risk prediction [22]. Banerjee and others compared various prediction methods, but their performance in capturing complex spatiotemporal dependencies and traffic network structures was limited [23].

Chen and others proposed the ESTGCN, but it had insufficient capabilities in handling traffic spatiotemporal relationships and accident prediction, especially in road network modeling [24]. Mekkaoui and colleagues introduced the DAS-GCN, which performs excellently in electric vehicle charging prediction but falls short in dynamic traffic risk prediction and spatiotemporal dependency modeling [25]. Chen proposed the STGCN-M, which has high accuracy in groundwater level prediction, but in spatiotemporal modeling for traffic network topologies and risk prediction, it may not be as flexible as hybrid algorithms [26].

Miao's method effectively evaluates dance movements but has weak capabilities in traffic risk prediction and complex network modeling [27]. Li proposed the CFRP-former framework, which has high accuracy in damage detection but performs weakly in spatiotemporal pattern and road network structure modeling for traffic risk prediction [28]. Suryanarayana's EOIQ-MC-STJGCN method performed well in malignant melanoma classification but fell short in traffic risk prediction and spatiotemporal dependency modeling [29]. Izadi and colleagues proposed a knowledge distillation-based ST-GNN

to optimize real-time traffic prediction, but it still has shortcomings in handling complex spatiotemporal dependencies and dynamic network topologies [30].

Yang and others' DEST-GNN performed well in photovoltaic power generation prediction but faced challenges in handling complex nonlinear spatiotemporal dependencies and multi-task learning [31]. Jeon and others combined CNN with STGCN to improve traffic speed prediction accuracy, but their model lacked interpretability and multi-task processing capabilities for complex spatiotemporal dynamics [32]. Tan's DCT-STGCN combined prediction method improved load forecasting for charging stations, but it performed limitedly in handling spatiotemporal dynamics and multi-task learning [33].

Ma and colleagues' STFGCN enhanced traffic prediction accuracy through spatiotemporal graph convolution operations, but its ability in handling complex spatiotemporal dependencies and cross-domain modeling remains insufficient [34]. Gao's ST-ENAGCN processed multi-ship distress identification, but there were limitations in complex navigation decision-making and data fusion [35]. Zhang and others' ASeer model performed better under dynamic traffic signals, but faced challenges in cross-domain generalization and handling multiple time-series data [36].

Chen and others' DSTGCN-based wind speed prediction model demonstrated strong robustness, but real-time performance and computational efficiency remain challenging [37]. Zeghina and others summarized spatiotemporal graph deep learning architectures applied across multiple fields, but the complexity of spatiotemporal data evolution and core problem solving remains insufficient [38].

With the continuous advancement of deep learning technologies, Convolutional Neural Networks (CNN) are used to extract spatial features, such as handling lane-changing distances, road networks, and other spatial data. Long Short-Term Memory Networks (LSTM) are widely used for sequence data modeling [39]. LSTM has significant advantages in processing time-series data and can capture long-term dependencies between data points. As a result, LSTM is extensively applied in fields such as speech recognition, natural language processing, and stock prediction. Graph Neural Networks (GNN) are used to process structured data from traffic road networks and model spatial dependencies between different road segments.

This study proposes a hybrid deep learning model framework based on CNN, LSTM, and GNN. The model uses cross-sectional data from multiple road segments, including highways, ring expressways, national roads, and urban roads. This approach directly extracts time-series and spatial-series features from raw vehicle location and speed data to build an accident probability mapping model. The study first designs a data collection scheme tailored to different traffic scenarios. Then, a series of filtering methods are applied to the collected data to ensure data quality. Finally, a CNN + LSTM + GNN hybrid deep learning network model is constructed to achieve end-to-end accident risk prediction in both temporal and spatial dimensions. The feasibility of the proposed method is validated through experiments conducted in complex scenarios.

## Data acquisition

The data collection for this study mainly focuses on four typical scenarios: highways, ring expressways, national roads, and urban roads. A total of 30 multi-dimensional sensor arrays were deployed on the Wuwei Expressway, a bi-directional 4-lane road, from Wuhan to Ezhou, with unidirectional deployment. Additionally, 25 sensors were deployed on the Huyu Expressway, a bi-directional 4-lane road near Huangshi City, connecting to the Dagang Expressway, with unidirectional deployment from Chongqing to Shanghai.Forty-five multi-dimensional sensors were deployed on the Wuhan Fourth Ring Road, a bi-directional 8-lane ring expressway, connecting from the Huangpi District of Wuhan to the Wuwei Expressway in the east, with unidirectional deployment. The above sensors collect real-time vehicle data, including speed, acceleration, lane-changing distance, and coordinates, at a frequency of 200Hz, gathering 200 data points per second. High-definition cameras were installed at the entrances and exits of highways and ring expressways to record vehicle trajectory videos at a frame rate of 50 frames per second. For the national road scenario, segments of G230, G316, and G318 around Wuhan were selected. In the urban road scenario, 15 main roads,

25 secondary roads, and 40 branch roads in Wuhan were randomly selected as the sampling framework. Due to the complex structure of national roads and urban roads, a stratified sampling strategy was employed. An observation point was set every kilometer on the selected roads, equipped with 4 millimeter-wave radars and panoramic cameras to collect relevant vehicle data at a frequency of 30Hz. Traffic flow monitoring devices were installed at major intersections to collect traffic flow data for model calibration. After approximately 100 days of data collection, around 379,000 vehicle trip trajectory data points were collected, with a total memory capacity of approximately 2.5TB. Detailed statistical information is provided in Table 1.

To visually present the vehicle trajectory features and driving operation data collected in this study, Figs 1–4 display the time series curves of speed, acceleration, lane-changing distance, and steering angle for five vehicles over a 5-minute period on a specific road segment. From the figures, it can be observed that there are significant individual differences in driving behaviors between vehicles.Speed and acceleration data exhibit typical high-frequency fluctuation characteristics under varying levels of noise interference. Therefore, it is necessary to perform smoothing and denoising on the raw trajectory data.

## Data preprocessing

To ensure the accuracy of the model's predictions, it is necessary to filter the collected vehicle trajectory data to eliminate outliers and noise. To achieve this, we have designed an outlier detection algorithm based on physical constraints and statistical rules. Data preprocessing is the prerequisite for outlier detection, and it involves the following steps:

**Table 1. overview of data collection.**

| Scenario | number of road sections/ observation points | acquisition equipment | Sampling frequency | vehicle sample size | Data volume |
|---|---|---|---|---|---|
| Highways | 55road sections | Multi-dimensional sensor array, high-definition camera | 200Hz, 50 fps | 125,828 vehicles | 0.83TB |
| Ring expressway | 45road sections | Multi-dimensional sensor array, high-definition camera | 200Hz, 50 fps | 128,860 vehicles | 0.85TB |
| National roads | 50 observation points | Millimeter-wave radar, panoramic camera, traffic flow monitor | 30Hz | 60,640 vehicles | 0.4TB |
| Urban roads | 45 observation points | Millimeter-wave radar, panoramic camera, traffic flow monitor | 30Hz | 63,672 vehicles | 0.42TB |

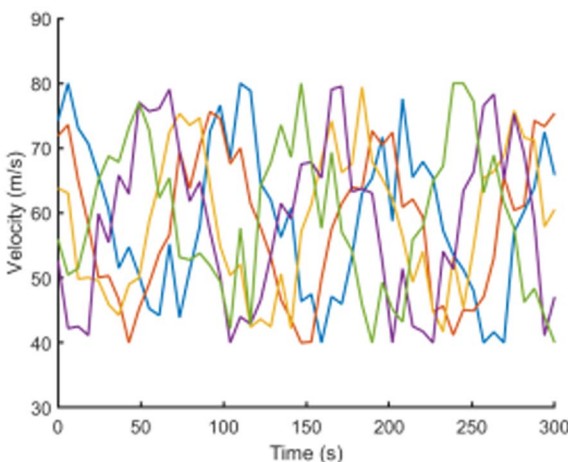

**Fig 1. Vehicle speed time series curves.**

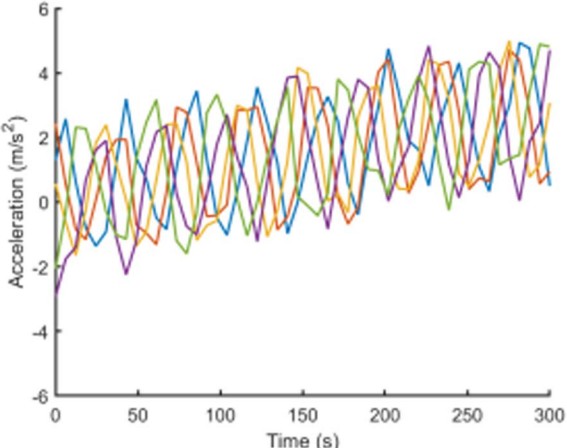

**Fig 2. Acceleration time series curves.**

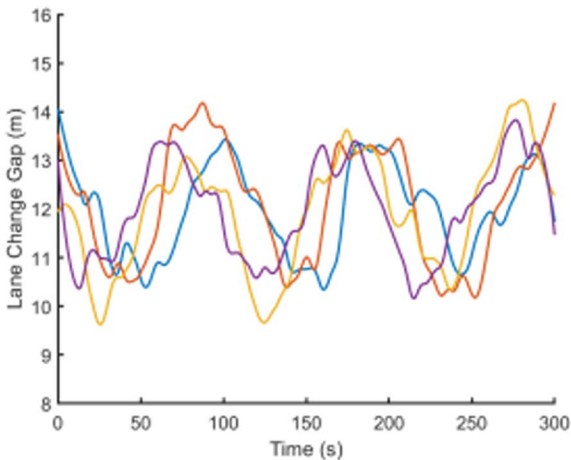

**Fig 3. Change lane spacing time series curves.**

First, invalid data, such as missing values and duplicate records, are removed. Next, physical quantities from different dimensions, such as speed, acceleration, lane-changing distance, and position, are standardized to facilitate subsequent comparisons and analysis. It is also ensured that the timestamps of each vehicle trajectory data are correct, and time alignment is performed to avoid errors caused by data synchronization issues.

In the vehicle trajectory data, abnormal data is judged based on the physical constraints of the traffic scenario. For example, the vehicle's speed is checked to seeif it falls within the road's speed limits. If the vehicle speed exceeds the maximum allowed speed or falls below the minimum speed at any given time, it is considered abnormal. The acceleration of the vehicle should generally be within a certain range. If it exceeds this range, abnormal data may be present. Assuming the normal acceleration range is ±12 m/s², any value outside this range is considered abnormal.

Additionally, the positions of two consecutive sampling points are checked to ensure they are consistent with the vehicle's speed. By calculating the expected position of the vehicle between two samples and comparing it with the actual sampled position, any significant deviation is classified as abnormal.

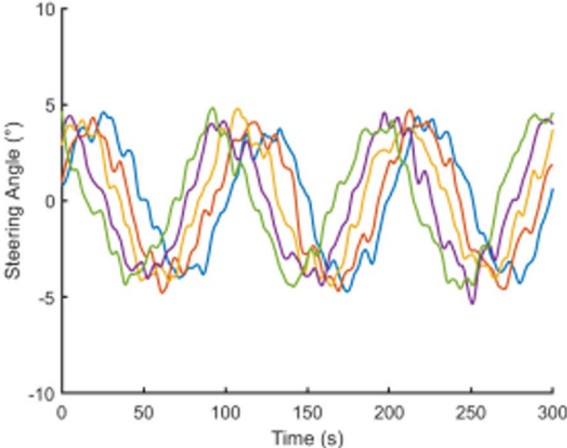

**Fig 4. Steering angle time series curve.**

In addition to the physical constraint-based detection, statistical methods are also needed to identify abnormal data. For each dimension of data (such as speed, acceleration, etc.), the mean and standard deviation are calculated. According to statistical principles, data points that exceed the mean ± 3 times the standard deviation are considered outliers.Box plots are used to detect extreme values in the data. The box plot method determines outliers by calculating the quartiles (Q1, Q3) and interquartile range (IQR). Data points that fall below Q1 - 1.5IQR or above Q3 + 1.5IQR are considered abnormal.Based on the above physical constraints and statistical rules, outlier data is finally filtered. For each data trajectory, both the physical constraints and statistical rules are applied to detect outliers. These outliers are removed, and the data within the normal range is retained.The optimized outlier detection algorithm reduces the proportion of outliers from 2.81% in the preliminary screening to below 0.29%, thereby improving the quality of the data used for model training.

To eliminate high-frequency random noise in vehicle sensor data, wavelet transform (WT) can be used for signal denoising. The wavelet threshold denoising algorithm is widely applied in signal processing, particularly for handling non-stationary signals such as time-series data. The wavelet transform-based denoising method effectively suppresses high-frequency noise while preserving the main features of the signal. The key idea behind the improved wavelet threshold denoising algorithm is to select an appropriate threshold to process the wavelet-transformed coefficients, thereby reducing the impact of noise on the signal. The signal is decomposed using wavelet transform, breaking the original signal into low-frequency and high-frequency components at different scales. The wavelet transform decomposes the signal into multiple scales of detail coefficients and approximation coefficients. For a given signal $x(t)$, applying wavelet transform results in several wavelet coefficients, typically represented as follows:

$$x(t) = \sum_i c_i \cdot \psi_i(t) \tag{1}$$

In this context, $c_i$ represents the wavelet coefficients, and $\psi_i(t)$ is the waveform function based on a specific wavelet basis.

To effectively preserve the details of the signal, this study adopts the soft thresholding method to process the wavelet coefficients. For each coefficient, if its absolute value is greater than the threshold, the threshold value is subtracted; if it is smaller than the threshold, it is set to zero. The formula is as follows:

$$\hat{c} = sign(c_i) \cdot \left(\max\left(|c_i| - T, 0\right)\right) \tag{2}$$

For dynamic signals such as vehicle sensor data, this study adopts an adaptive threshold based on noise estimation. Noise is typically reflected in the high-frequency components, and the threshold is dynamically selected by estimating the standard deviation of the high-frequency coefficients. The standard deviation of the high-frequency coefficients estimates the intensity of the noise, denoted as $\sigma$, and the estimation method is as follows:

$$\sigma = \frac{1}{\sqrt{2}} \cdot median\left(|c_i|\right) \tag{3}$$

The adaptive threshold method is given by

$$T = \sigma \cdot \sqrt{2\log(N)} \tag{4}$$

where $\sigma$ represents the noise estimation and $N$ is the signal length.

After the denoising process, the denoised coefficients are reconstructed back into the original signal's time series using the inverse wavelet transform. The formula is as follows:

$$\hat{x}(t) = \sum_i \hat{c}_i \cdot \psi_i(t) \tag{5}$$

Here, $\hat{c}$ represents the denoised coefficients, and $\psi_i(t)$ is the wavelet basis function.

This improved wavelet threshold denoising algorithm better preserves the main features of the signal while effectively suppressing noise, thereby retaining the key characteristics of vehicle movement. Fig 5 shows a comparison of vehicle speed before and after denoising.

To fill in the missing values in vehicle trajectory data, this study proposes an adaptive spatiotemporal weighted interpolation strategy. This method assigns different weights to known data points based on their temporal and spatial distances. Data points that are closer in time and space are given higher weights.The interpolation result is calculated using a weighted average, and the weight function parameters are adaptively adjusted based on the distribution of missing data. This strategy effectively considers the spatiotemporal dependencies of traffic data, improving interpolation accuracy. It ensures that the filling of missing data aligns with the physical consistency of the original data, thus providing high-quality data support for the traffic accident risk prediction model.

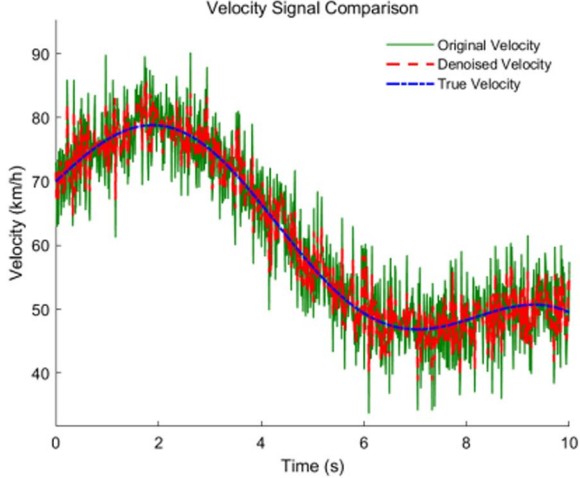

**Fig 5. Effect of wavelet thresholding denoising.**

To eliminate the influence of different physical dimensions on subsequent analysis, this study uses the Z-score normalization method to standardize the multidimensional time series and spatial sequences of vehicle speed and acceleration, as shown in Fig 6. Specifically, the mean and standard deviation of each feature are first calculated, and then each data point is standardized using the following formula:

$$z = (x - \mu) / \theta \qquad (6)$$

Where $x$ is the original data, $\mu$ is the mean of the feature, and $\theta$ is the standard deviation of the feature.

Through standardization, the mean of all feature values is set to 0, and the standard deviation is set to 1. This eliminates the dimensional differences between different features. As a result, the model can converge better during training and improve prediction accuracy. This method effectively avoids the impact of large or small feature values on model training. It enhances the robustness and generalization ability of the model.

## Model selection

Compared to the methods mentioned earlier, as shown in Table 2, the CNN+LSTM+GNN combination algorithm performs excellently in handling spatiotemporal features, prediction accuracy, robustness, and computational efficiency. By combining CNN to extract local spatial features, LSTM to capture temporal features, and GNN to model global spatial dependencies, the combined algorithm is able to more comprehensively capture the complex patterns of traffic flow and accident risk.

Traffic accident risk prediction is a typical classification problem involving both time series and spatial sequences. In this study, we compared the applicability of mainstream deep learning models, such as PSO-LSTM, CDBL, DHSTNet, STD-Net, and the CNN + LSTM + GNN combination model, for this problem.

To quantitatively compare the performance of these models in real-world scenarios, we designed a set of prediction performance evaluation experiments. The experiments were conducted on a Windows 11 system, using the Python programming language and the PyTorch deep learning framework. The hardware setup included an NVIDIA GPU, 16GB of memory, and 1TB of storage. We used trajectory data from highways, ring expressways, national roads, and city roads, with vehicle speed, acceleration, lane change distance, and location as inputs. We trained five algorithm models with

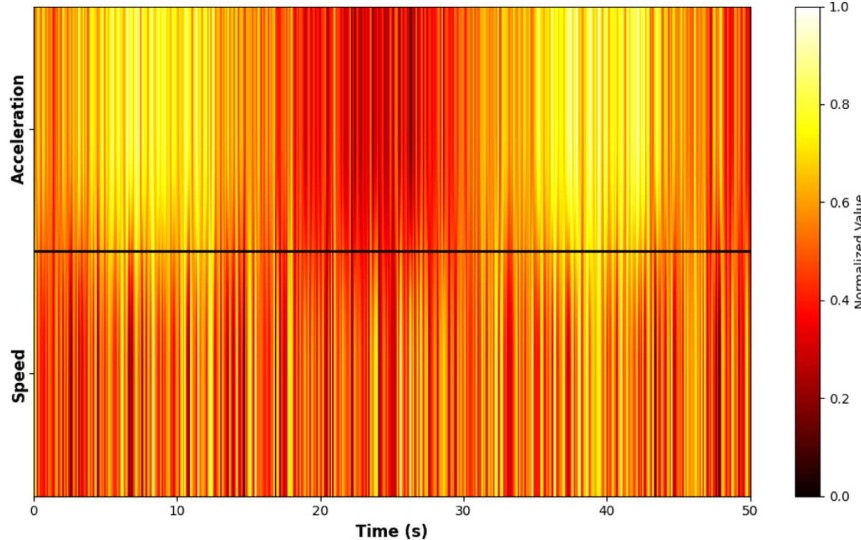

**Fig 6. Heatmap of Normalized Features.**

**Table 2.  Comparison of Traffic Accident Risk Prediction Models Based on Different Algorithms.**

| Algorithm | Spatial Feature Extraction | Temporal Feature Extraction | Global Spatio-temporal Dependency Capture | Robust-ness |
|---|---|---|---|---|
| Wang and Shi's SVM + Wavelet Analysis | Distance-based spatial feature extraction | Short-term speed prediction | None | Medium |
| Qing's Random Forest Model | Distance-based spatial feature extraction | Temporal segmentation feature extraction | None | Medium |
| Wenhua's K-means + PSO Optimization Model | K-means based spatial feature extraction | PSO optimized temporal feature extraction | None | Medium |
| Senyin's Radar Detector Model | Radar-based spatial feature extraction | Spatiotemporal classification feature extraction | None | Moder-ate |
| Pan's PSO-LSTM Model | PSO optimized spatial feature extraction | LSTM-based temporal feature extraction | None | Good |
| Li's CDBL Framework | Behavior learning-based spatial feature extraction | Continuous behavior learning temporal feature extraction | Moderate | High |
| Ali's DHSTNet Model | Spatiotemporal neural network feature extraction | Spatiotemporal neural network temporal feature extraction | Good | High |
| Li's STEGCN Model | Graph convolution-based spatial feature extraction | 1D convolution-based temporal feature extraction | Moderate | High |
| Liu's RT-GCN Model | Graph convolution-based spatial feature extraction | Attention mechanism-based temporal feature extraction | Moderate | High |
| Fan's RGDAN Model | Graph diffusion-based spatial feature extraction | Time attention-based temporal feature extraction | Moderate | High |
| CNN+LSTM+GNN Hybrid Model | CNN extracts spatial features, LSTM extracts temporal features, GNN captures global spatial dependencies | LSTM captures temporal dependencies, CNN and GNN jointly optimize | Very High | Very High |

binary classification labels indicating whether a vehicle will have an accident within the next 5 seconds.Through 10-fold cross-validation, we obtained the accuracy, precision, recall, specificity, mean squared error (MSE), mean absolute error (MAE), area under the curve (AUC) (ROC curve), and F1 score for each model. Accuracy measures the proportion of correct predictions, and it is suitable for balanced classes. Precision focuses on the accuracy of positive predictions, reducing false positives. Recall emphasizes the completeness of positive predictions, reducing false negatives. Specificity focuses on the accuracy of negative predictions, reducing false positives. For regression tasks, MSE and MAE assess the model's prediction errors, with MSE being more sensitive to large errors, while MAE is more intuitive. The AUC value reflects the model's ability to distinguish between positive and negative cases, which is especially important in imbalanced classes. The F1 score combines precision and recall, balancing accuracy and completeness, providing a comprehensive measure of model performance.

Different metrics emphasize different aspects in various scenarios. By considering these metrics comprehensively, we can better understand the strengths and weaknesses of the models. As shown in Table 3, the data represents the average values from multiple experiments, with real values labeled later. The CNN+LSTM+GNN model demonstrates significant advantages across several key metrics. Its accuracy reached 94.5%, the highest among all models, indicating superior overall prediction accuracy. Its precision was 94.0%, showing better performance in reducing false positives. The recall rate was 93.7%, reflecting the model's ability to capture real accident risks and effectively reduce the miss rate. Specificity was as high as 95.8%, demonstrating that the model can maintain high risk detection efficiency while avoiding false positives. Additionally, the model achieved MSE and MAE values of 0.030 and 0.063, respectively, both lower than those of other models, showcasing its excellent performance in spatiotemporal feature modeling and prediction accuracy. The AUC value was 0.978, indicating outstanding risk differentiation capability. The F1 score was 93.8%, showing a good balance between precision and recall.

**Table 3. Comparison of the predictive performance of candidate models.**

| Model | Accuracy | Precision | Recall | Specificity | Mean Squared Error (MSE) | Mean Absolute Error (MAE) | AUC (ROC Curve) | F1-Score |
|---|---|---|---|---|---|---|---|---|
| PSO-LSTM | 89.2% [88.9%, 89.6%] | 88.5% [88.2%, 88.9%] | 86.8% [86.4%,87.2%] | 90.3% [89.9%, 90.6%] | 0.045 [0.041, 0.047] | 0.089 [0.083, 0.091] | 0.923 [0.921, 0.927] | 87.6 [87.3, 88.0] |
| CDBL | 90.1% [89.8%, 90.6%] | 89.8% [89.3%, 90.3%] | 87.9% [87.4%,88.3%] | 91.2% [90.9%, 91.5%] | 0.041 [0.038, 0.043] | 0.082 [0.079, 0.086] | 0.937 [0.933, 0.941] | 88.8 [88.3, 89.4] |
| DHSTNet | 91.0% [90.6%, 91.3%] | 90.5% [90.1%, 90.9%] | 89.3% [88.9%,89.6%] | 92.1% [91.8%, 92.4%] | 0.038 [0.035, 0.041] | 0.078 [0.073, 0.082] | 0.945 [0.941, 0.947] | 89.9 [89.4, 90.3] |
| STD-Net | 92.3% [91.9%, 92.7%] | 91.8% [91.5%, 92.2%] | 90.7% [90.3%,91.2%] | 93.2% [92.8%, 93.6%] | 0.035 [0.033, 0.039] | 0.072 [0.069, 0.075] | 0.953 [0.949, 0.955] | 91.2 [90.9, 91.6] |
| CNN+LSTM+GNN | 94.5% [94.1%, 94.7%] | 94.0% [93.6%, 94.4%] | 93.7% [93.3%, 94.1%] | 95.8% [95.3%, 96.2%] | 0.030 [0.026, 0.033] | 0.063 [0.059, 0.067] | 0.978 [0.973, 0.981] | 93.8 [93.3, 94.1] |

By integrating CNN for spatial feature extraction, LSTM for capturing temporal dependencies, and GNN for modeling the complex structure of traffic networks, the CNN+LSTM+GNN model can more comprehensively handle diverse traffic scenarios. Its strong generalization ability and stability make it the optimal solution for predicting traffic accident risks. In contrast, other models such as PSO-LSTM, CDBL, DHSTNet, and STD-Net show some improvement in individual metrics but do not outperform the combination model in terms of overall performance and generalization ability.

In conclusion, the performance of the CNN+LSTM+GNN model fully validates its superiority and application potential in traffic accident risk prediction. Therefore, this study decided to adopt the CNN + LSTM + GNN combination model for modeling.

## Model building

To construct a deep learning-based vehicle trajectory traffic accident risk prediction model, we integrate the features of CNN, LSTM, and GNN to fully leverage their respective strengths, thereby enhancing the model's predictive performance. The input-output flow of the model is shown in Fig 7. To ensure that the model can handle spatiotemporal features, we need to prepare and organize the input data.

CNN Input Data: For spatial features (such as vehicle speed, acceleration, lane change distance, etc.), CNN receives data with spatial structure. The vehicle features at each time point (e.g., speed, acceleration, lane change distance) form a two-dimensional matrix with the shape of $[N,H,W,C]$, where $N$ is the number of samples, $H$ and $W$ are the height and width of the feature map, respectively, and $C$ is the number of channels. For example, for the three features of speed, acceleration, and lane change distance, three channels can be used.

LSTM Input Data: LSTM receives time series data, primarily processing the vehicle's motion features at different time steps, such as speed, acceleration, etc. The input data has the shape of $[N,T,F]$, where $N$ represents the number of samples, $T$ denotes the number of time steps, and $F$ is the feature dimension at each time step. LSTM aims to capture the temporal dependencies in the vehicle trajectory data.

GNN Input Data: GNN processes traffic network structural data, where node features represent segment information (such as traffic flow, segment length, traffic conditions, etc.), and the adjacency matrix represents the relationships between segments. The input data includes the node feature matrix and the adjacency matrix, which respectively represent the attributes of each segment and the spatial dependencies between segments.

For the input data of each model, we separately extract spatial features, temporal features, and traffic network structural features, as shown in Fig 8, which displays heatmaps of the time distribution of features such as vehicle speed, acceleration, and lane change distance.

CNN Feature Extraction: CNN processes spatial features (such as vehicle speed, acceleration, etc.) through convolutional layers. First, local spatial features are extracted using convolutional layers, followed by dimensionality reduction

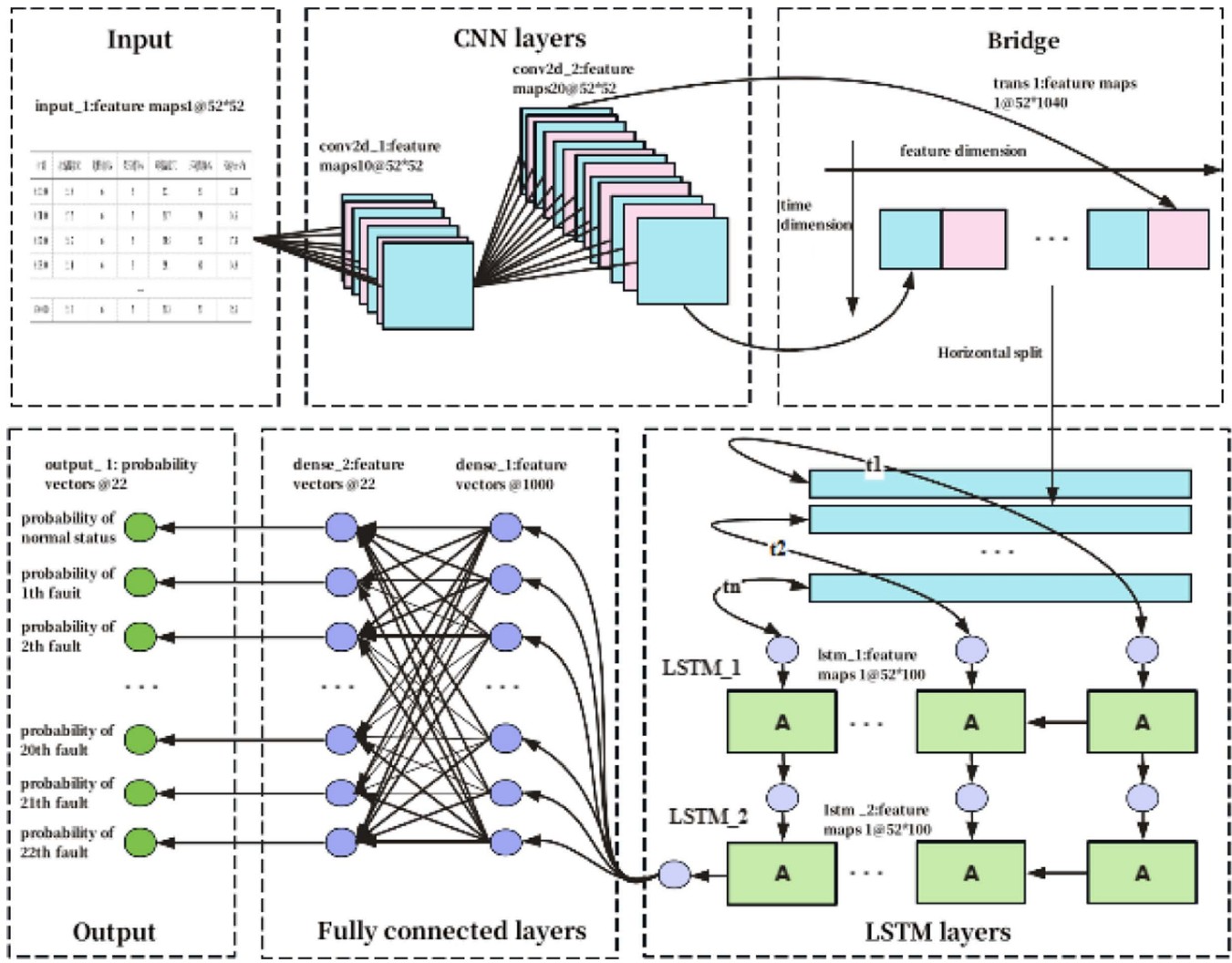

**Fig 7. Structure of CNN + LSTM + GNN accident risk prediction model.**

through pooling layers. The features are then passed to the fully connected layers for further mapping. The output of CNN is a vector representing the spatial features.

LSTM Feature Extraction: LSTM effectively captures long-term dependencies in time series data through its gating mechanisms. The output of LSTM is the hidden state at each time step, and we select the hidden state of the final time step as the temporal feature representation of the vehicle trajectory.

GNN Feature Extraction: GNN processes the spatial features of the road network through graph convolutional layers, learning the representation of each node. GNN is capable of capturing the spatial dependencies between road segments and outputs the feature representation for each road segment.

After feature extraction, we need to fuse the features extracted by CNN, LSTM, and GNN to integrate the advantages of all three models. We use feature concatenation, where the output vectors of the three models are concatenated along the feature dimension to obtain a unified feature representation. The spatial features extracted by the CNN network are a vector with dimension $D_{cnn}$, the time-series features extracted by LSTM are a vector with dimension $D_{lstm}$, and the road

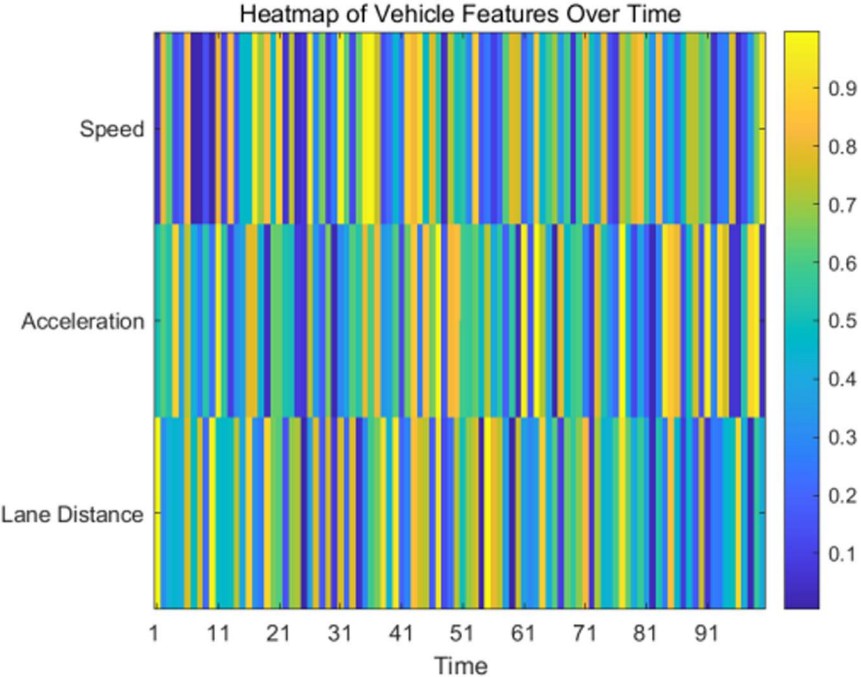

**Fig 8. Heatmap of Normalized Features Over Time.**

network spatial features extracted by GNN are a vector with dimension $D_{gnn}$. These three vectors are concatenated into a new feature vector using the torch.cat() function, and the concatenated feature vector has dimension:

$$D_{fusion} = D_{cnn} + D_{lstm} + D_{gnn} \tag{7}$$

The concatenated feature vector is processed through a Fully Connected Layer (FC) to perform dimensional transformation and nonlinear mapping. The fully connected layer maps the concatenated feature vector to a new feature space, resulting in a feature representation with higher expressive power. After the fully connected layer, the activation function ReLU (Rectified Linear Unit) is applied to enhance the network's ability to model nonlinear relationships. The output of the fully connected layer is used for the final accident risk prediction. The definition of ReLU is:

$$f(x) = \max(0, x) \tag{8}$$

When the input is negative, the output is 0; when the input is positive, the output is the input value itself. This effectively mitigates the vanishing gradient problem, thereby accelerating the training process. After these feature extraction and input steps, the model's output is a high-dimensional feature vector $h$, which contains useful information extracted from the spatiotemporal features. The extracted high-dimensional feature $h$ is then passed to one or more Fully Connected Layers (FC). The output of the Fully Connected Layer is a real-valued scalar $z$, that is:

$$z = Wh + b \tag{9}$$

Where A is the weight matrix of the Fully Connected Layer, and B is the bias term.

The Fully Connected Layer serves the purpose of feature transformation and dimensionality reduction, preparing the data for the final classification task. The output of the Fully Connected Layer is mapped to an accident risk probability

---

ranging from 0 to 1 using the sigmoid activation function. If the output is close to 1, it indicates a high probability of an accident occurring, whereas an output close to 0 suggests a low probability. This probability value represents the likelihood of a traffic accident occurring within the next 5 seconds. The processed output $z$ from the Fully Connected Layer is passed through the sigmoid activation function to produce the predicted accident probability $p$. The formula for the sigmoid activation function is as follows:

$$p = \sigma(z) = 1/\left(1 + e^{-z}\right) \tag{10}$$

which can also be expressed as:

$$p = \sigma(W \cdot h + b) \tag{11}$$

For the binary classification task, this study uses binary cross-entropy as the loss function:

$$Loss = -\frac{1}{n}\sum_{i=1}^{N}\left[y_i \log p_i + (1 - y_i)\log(1 - p_i)\right] \tag{12}$$

Where $n$ represents the number of samples, $y_i$ is the true label of the $i$ sample (0 or 1), and $p_i$ is the model's predicted probability of an accident occurring for the $i$ sample (output after applying the sigmoid activation function).

The model is trained using the Stochastic Gradient Descent with Momentum (SGD with Momentum) optimization algorithm. The model parameters $\theta_0$ are initialized with small random values $\theta_0 \sim N(0, \sigma^2)$, the learning rate is set to $\alpha = 0.01$, and the momentum hyperparameter is $\beta = 0.9$. $t$ represents the current iteration number, starting from 0, and the initial momentum $v_0 = 0$ is used to initialize the momentum for each parameter. The gradient of the loss function with respect to each parameter is then calculated:

$$\frac{\partial Loss}{\partial \theta_j} = \frac{1}{N}\sum_{i=1}^{N}(p_i - y_i)x_{ij} \tag{13}$$

Here, $x_{ij}$ represents the input feature of the $j$ parameter for the $i$ sample. Momentum is used to update the gradient information, and the update formula for momentum is as follows:

$$v_{t+1} = \beta v_t + (1 - \beta)\nabla_\theta J(\theta_t) \tag{14}$$

Here, $v_t$ represents the momentum, $\beta = 0.9$ is the momentum decay factor, and $\nabla_\theta J(\theta_t)$ is the gradient at step $t$. After each iteration, the parameters are updated using momentum, as described by the following formula:

$$\theta_{t+1} = \theta_t - \alpha v_{t+1} \tag{15}$$

Here, $\theta_t$ represents the current parameter, $\alpha$ is the learning rate, and $v_{t+1}$ is the updated momentum. This process is repeated until convergence (Table 4).

This study proposes a traffic accident risk prediction method based on a combination of CNN, LSTM, and GNN by deeply exploring the spatiotemporal features in vehicle trajectory data. To fully utilize the latent risk information contained in the vehicle trajectory data, we extracted motion features of the vehicles from multiple perspectives. These features include the average speed, average acceleration, the standard deviation of speed and acceleration, as well as steering angles within a sliding window.

Additionally, this study particularly focuses on high-risk events, such as rapid acceleration, rapid deceleration, and sharp turns, which are often associated with an increased risk of accidents. By combining the strengths of CNN, LSTM,

**Table 4. Notation.**

| Symbol | Description |
|---|---|
| $c_i$ | wavelet coefficients |
| $\psi_i(t)$ | wavelet basis function |
| $\hat{c}$ | denoised coefficients |
| $T$ | adaptive threshold |
| | noise estimation |
| $\sigma$ | signal length |
| $N$ | raw data |
| $x$ | mean of the feature |
| $\mu$ | standard deviation of the feature |
| $\theta$ | weight matrix of the fully connected layer |
| $W$ | high-dimensional feature vector |
| $h$ | sample size |
| $n$ | bias term |
| $b$ | true label of the i-th sample |
| $y_i$ | the model's predicted probability of an accident occurring for the i-th sample |
| $p_i$ | learning rate |
| $\alpha$ | momentum hyperparameter |
| $\beta$ | current iteration number |
| $t$ | input feature of the j-th parameter for the i-th sample |
| $x_{ij}$ | momentum |
| $v_t$ | gradient at step t |
| $\nabla_\theta J(\theta_t)$ | currently updated parameters |
| $\theta_t$ | updated momentum |

and GNN, this study is able to capture spatial features, temporal dependencies, and spatial dependencies in the traffic network, thus more accurately assessing the risk level of accidents.

Ultimately, the proposed model has demonstrated its outstanding prediction performance in various traffic scenarios. It provides effective accident early-warning technical support for intelligent transportation systems.

## Model training

This study uses a stratified sampling method to divide the dataset into training, validation, and test sets in an 8:1:1 ratio. The training set is used for model parameter learning, the validation set for hyperparameter tuning, and the test set for evaluating the model's generalization ability.

To address the class imbalance issue, an oversampling method is employed by randomly duplicating minority class samples (accident trajectory data). The number of these samples is increased to be similar to that of the majority class samples. The process of sample duplication involves randomly selecting samples from the minority class and repeatedly adding them to the training set to enhance the representation of accident data.

In addition to the basic duplication method, this study also applies the SMOTE (Synthetic Minority Over-sampling Technique) algorithm to generate new minority class samples. The SMOTE algorithm creates new synthetic samples by performing linear interpolation between minority class samples. This helps to increase the diversity of the minority class samples and prevent simple oversampling from causing model overfitting.

Specifically, SMOTE first selects a sample from the minority class and finds its k nearest neighbors. Then, it generates new synthetic samples through interpolation between the selected sample and its neighbors. During the oversampling process, to avoid generating an excessive number of minority class samples that would lead to data imbalance, this study controls the oversampling ratio. The specific ratio is determined through cross-validation to balance the ratio of positive and negative samples in the training set, while ensuring the effectiveness of model training.

After applying the oversampling strategy, the number of accident and non-accident samples in the training set is effectively balanced. The detailed configuration of the dataset is shown in Table 5. This approach allows the model to learn more accident-related features during the training phase, improving its prediction ability for minority class samples and enhancing the model's robustness when handling imbalanced data.

In terms of model hyperparameter selection, both model performance and computational efficiency were considered. The deep learning model designed in this study includes three main components: CNN, LSTM, and GNN.

First, the CNN section consists of three convolutional layers. The first layer has 64 filters, the second layer has 128 filters, and the third layer has 256 filters to enhance feature extraction capability. The kernel size is set to 5x5, and a max-pooling layer is used for dimensionality reduction. The pooling kernel size is 2x2 with a stride of 2. The dropout rate is set to 0.4 to prevent overfitting. Batch normalization is applied after each convolutional layer to accelerate training and improve stability. The ReLU activation function is added after the convolution layers.

The LSTM section consists of two LSTM layers, each with 256 LSTM units, and a hidden state dimension of 256. This design aims to capture long-range dependencies in the time series. The dropout rate is set to 0.3, and the activation function is tanh. Batch normalization is applied between the LSTM layers to reduce overfitting and stabilize the training process. A bidirectional LSTM (BiLSTM) is used to enhance the model's ability to capture both past and future temporal information.

The GNN section consists of three layers, with the ReLU activation function to capture spatial dependencies in the traffic network. The dropout rate is 0.4, and batch normalization is applied after each GNN layer to improve model stability and training efficiency. To enhance graph convolution performance, GCN (Graph Convolutional Network) is used as the core architecture for the GNN.

The fully connected layers are set to two layers. The first layer has 512 neurons, and the second layer has 256 neurons. ReLU is used as the activation function, and a dropout layer (with a dropout rate of 0.5) is added to further reduce overfitting. Between the fully connected layers, L2 regularization (weight decay) is applied to prevent overfitting. The output layer consists of one neuron with a sigmoid activation function, suitable for binary classification tasks.

The initial learning rate is set to 0.001, using the Adam optimizer and an exponential decay learning rate strategy. During training, the F1 score is evaluated every 10 epochs. If the F1 score on the validation set does not improve for five consecutive cycles, early stopping is triggered, and the best model is saved. The batch size is set to 128, and the number

**Table 5. Division of the data set.**

| Data set | Sample size | Proportion of accident sample | Proportion of accident samples after oversampling |
|---|---|---|---|
| Training set | 303,200 | 4.8% | 9.7% |
| validation set | 37,900 | 4.7% | 9.8% |
| Test set | 37,900 | 4.5% | 9.5% |

of training epochs is set to 180. Gradient clipping is used during training to prevent gradient explosion, ensuring stable training.

Additionally, data augmentation techniques are employed to generate more training samples by adding random noise and rotation, further improving the model's robustness and generalization ability.

Fig 9 shows the model's training curve. After approximately 60 iterations, the loss function converges, and the F1 score on the validation set reaches a stable level. This figure illustrates vehicle trajectory prediction and traffic accident risk analysis based on deep learning and spatiotemporal features.

Fig 10 shows the connection between the vehicle's predicted trajectory and risk prediction. Each trajectory represents a vehicle's actual position changes (solid line) over time and its predicted future position changes (dashed line) based on the deep learning model. Data for 10 vehicles are presented. By comparing actual and predicted trajectories, the model's prediction effectiveness can be intuitively assessed, validating its ability to predict vehicle trajectories.

The color depth of the trajectories reflects the accident risk level at different time points during the vehicle's journey. The gradient color from green to red indicates a transition from low risk to high risk. The risk level is assessed based on the vehicle's motion state, speed, acceleration, and other spatiotemporal features. As the vehicle's position changes, the accident risk fluctuates. The color change helps us better understand the spatiotemporal distribution of risks.

This demonstrates that the constructed CNN + LSTM + GNN model can effectively learn accident risk features from vehicle trajectory data, providing a foundation for subsequent pattern recognition and early warning analysis in academic research writing.

In the training samples, Fig 11 shows the statistical distribution of positive and negative samples. From the figure, it can be observed that in the positive samples, where accidents occur, the standard deviations of speed, acceleration, and the number of rapid decelerations are generally higher than those in the negative samples, where no accidents occur. This phenomenon confirms the correlation between these indicators and accident risk.

Additionally, it is observed that different indicators have varying abilities to distinguish between sample types. For example, the distribution of average speed does not show as clear a distinction between the two sample types as the

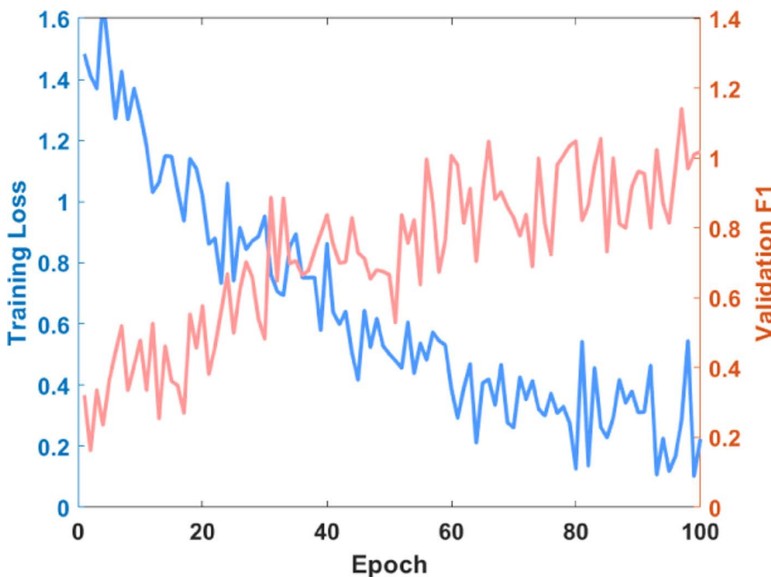

**Fig 9. Model training curve.**

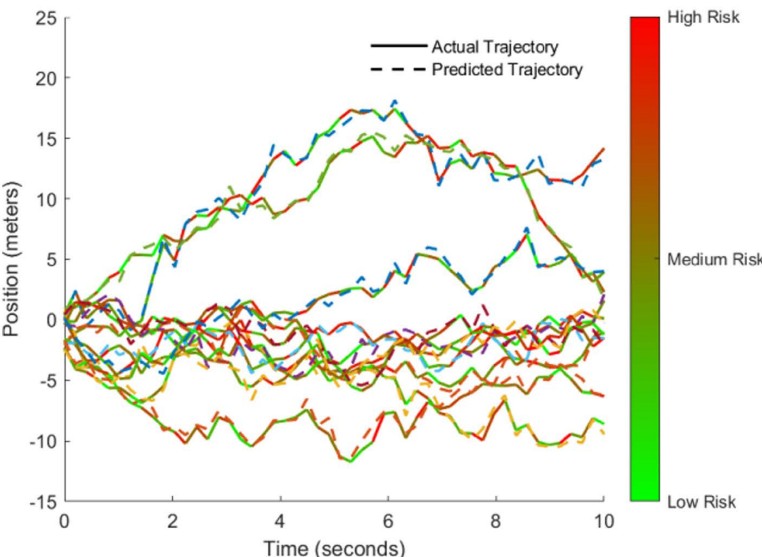

**Fig 10. Time vs. Position Trajectory Prediction with Risk Levels.**

standard deviation of speed. Therefore, to fully leverage the predictive potential of each feature, it is necessary to assign appropriate weights to different features through an end-to-end training process.

## Statistical analysis of data

The vehicle trajectory data collected in this study includes multiple key indicators such as vehicle speed, acceleration, steering angle, and lane change distance. To thoroughly analyze the distribution characteristics of these indicators, a detailed statistical analysis was conducted. Fig 12 shows the probability density curves of different indicators, with data representing the average values from multiple statistics. Standard deviation bars are added to indicate the range of data variation in the experiment.

The analysis results show that vehicle speed follows a Gaussian distribution, with an average speed of 55.3 km/h and a standard deviation of 18.7 km/h. The acceleration data exhibits a clear Laplace distribution, with the peak concentrated at 0 m/s². This characteristic is closely related to the frequent starting and stopping, as well as acceleration and deceleration behaviors of the vehicle. The distribution of the steering angle is relatively uniform, without showing significant bias. The lane change distance distribution follows a stable gamma distribution, suggesting that vehicles generally maintain a fixed safety distance when changing lanes.

Additionally, high-risk events were labeled based on thresholds for speed, acceleration, steering angle, and lane change distance. The experimental results show significant differences in the proportion of high-risk events across different road scenarios. In the highway scenario, approximately 7.2% of the data points were identified as high-risk events. In urban roads, this proportion increased to 14.5%. Further analysis revealed that the proportion of high-risk events was 9.1% in the national road scenario and 8.3% in the ring highway scenario. Lane change distance had a significant impact in these scenarios, especially in urban roads and national roads, where the probability of high-risk events increased significantly when the lane change distance was shorter.

These differences reflect the influence of different road types and traffic environments on driving risk. The complex traffic environment in urban roads significantly increases driving risk, while the risks on national roads and ring highways are lower than those on urban roads but higher than on traditional highways. Based on this analysis, this study adopted

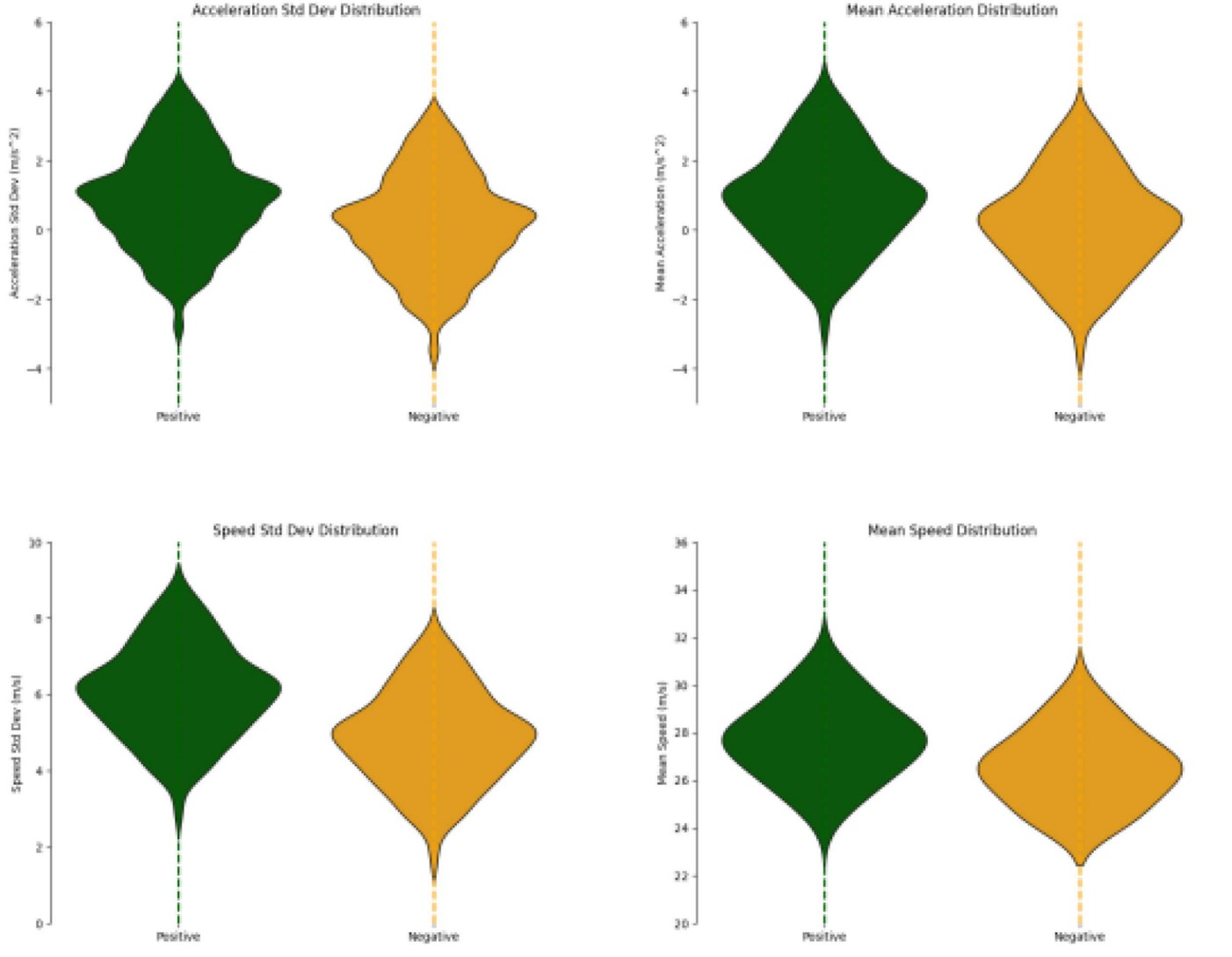

**Fig 11. Comparison of the Distribution of Statistical Indicators for Positive and Negative Samples.**

a combination model (including multiple indicators such as speed, acceleration, steering angle, and lane change distance) to further enhance the prediction accuracy of high-risk events. By comprehensively evaluating these features, the model can more effectively identify potential high-risk driving behaviors and provide decision support for traffic safety management.

## Prediction precision and error analysis

To comprehensively evaluate the performance of the CNN+LSTM+GNN combined model in the accident risk prediction task, several metrics including precision, recall, F1 score, specificity, accuracy, and MCC (Matthews Correlation Coefficient) were used for quantitative analysis. The confusion matrix for the model's predictions on the test set is shown in Table 6. The model accurately identified 4910 out of 5324 positive samples, achieving a recall rate of 92%. It correctly classified 5889 out of 6232 negative samples, with an accuracy rate of 93.5%. The specificity is 95.8%, and the MCC is

0.875, indicating that the model achieves a balanced performance across both types of samples. The AUC value of the model's ROC curve is as high as 0.965, further demonstrating the model's strong discriminative ability.

However, some typical errors were also observed. For example, the model occasionally misclassified a normal lane change on a biased lane as a dangerous event. This suggests that incorporating environmental information such as lane markings and traffic signals in future research could improve the model's adaptability to different scenarios.

## Generalization ability and robustness test

To verify the generalization ability of the constructed model, its prediction performance was evaluated in different test scenarios. Fig 13 compares the model's precision, recall, F1 score, specificity, and other metrics across four types of roads: highway, ring road, national road, and city roads. It can be observed that, despite significant differences in data distribution across scenarios, the model consistently maintains high and stable prediction performance. The variation in the F1 score is less than 3%, indicating that the CNN+LSTM+GNN-based accident risk prediction method has good environmental

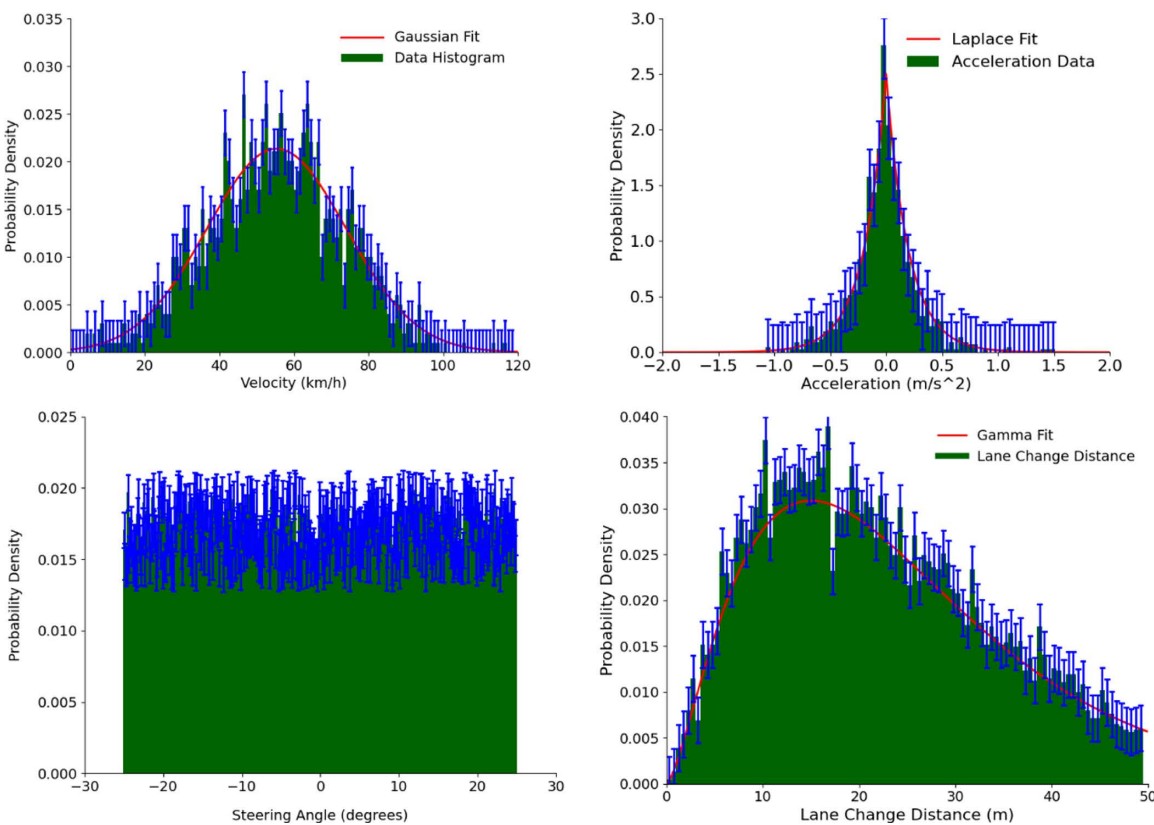

**Fig 12. Distribution of key indicators of vehicle trajectory.**

**Table 6. Confusion matrix for model prediction.**

| real\forecast | Standard practice | negative example |
|---|---|---|
| cases (accidents) | 4910 | 414 |
| negative examples (security) | 343 | 5889 |

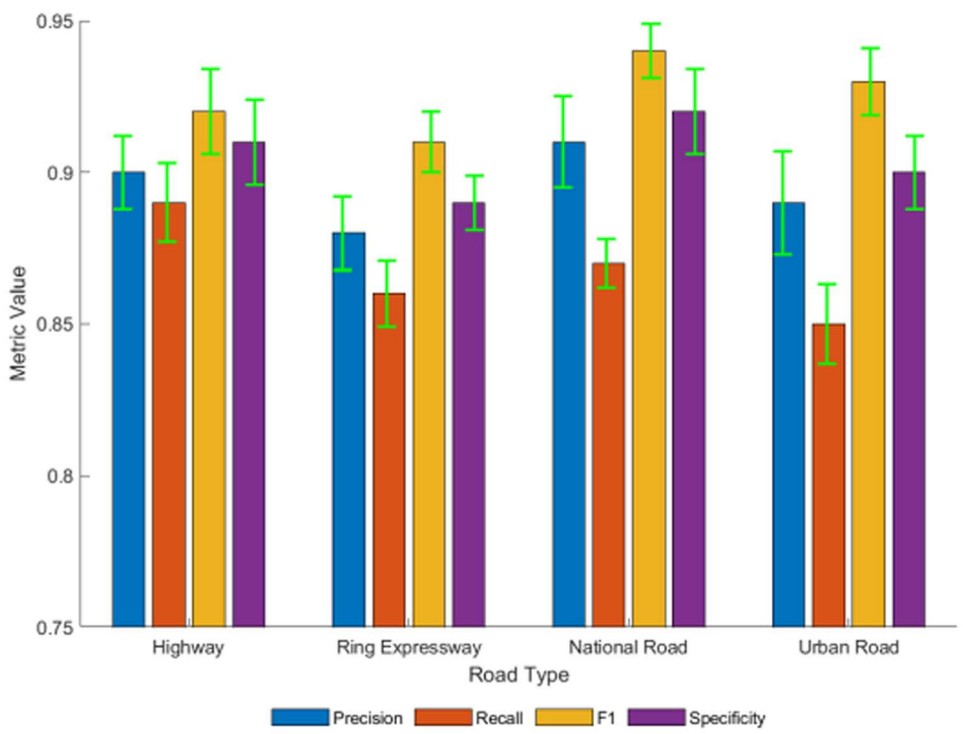

**Fig 13. Predictive performance of the model in different scenarios.**

adaptability. At the same time, the model's prediction performance remains stable under different vehicle types, weather conditions, and driving behaviors, demonstrating strong robustness.

To evaluate the performance of the proposed CNN+LSTM+GNN combination model in large-scale data processing, multiple experiments were conducted. Tests were performed on a server equipped with an NVIDIA Tesla V100 GPU using datasets of different sizes. The results show that as the data volume increases, the model's training time grows significantly, but its prediction speed remains stable, processing approximately 6000–7000 data points per second. This indicates that the model is fully capable of meeting the needs of most real-time traffic monitoring systems.

Additionally, we tested the model's deployment performance on an edge device (NVIDIA Jetson AGX Orin). The results revealed that the model could process approximately 1500 data points per second, a performance level sufficient to support real-time accident risk assessment for a single intersection or a small section of road.

To further optimize performance, we introduced model quantization, reducing the model's precision from FP32 to INT8. With this adjustment, the inference speed increased by about 35%, while memory usage and power consumption were significantly reduced, with the prediction accuracy remaining largely unchanged.

Experiments also analyzed the impact of different batch sizes on training and inference performance. The results showed that a batch size of 128 achieved a good balance between training efficiency, model performance, and memory usage. During inference, increasing the batch size further enhanced device utilization and accelerated overall inference speed.

These results demonstrate that the proposed CNN+LSTM+GNN combination model not only has outstanding prediction accuracy and generalization ability but also has significant application potential in large-scale real-time systems, as shown in Table 7.

**Table 7. Detailed comparison of CNN+LSTM+GNN model performance and resource consumption.**

| volume of data (10,000 pieces) | Training time (hours) | Predicted speed(bars/second) | GPU memory footprint (GB) | Model size (MB) | FP32 inference time (ms) | INT8 inference time (ms) | Batch size |
|---|---|---|---|---|---|---|---|
| 10 | 2.1 | 7000 | 3.5 | 65 | 0.9 | 0.7 | 32 |
| 50 | 8.3 | 6800 | 6.1 | 65 | 0.95 | 0.75 | 64 |
| 100 | 14.5 | 6600 | 8.4 | 65 | 1.0 | 0.8 | 128 |
| 500 | 61.7 | 6400 | 11.9 | 65 | 1.05 | 0.85 | 256 |

Fig 14 illustrates the relationship between dataset size, training time, prediction accuracy, and memory usage. As the dataset size increases, both training time and memory usage show an upward trend, indicating that as the data volume grows, the computational and storage demands of the model also increase. However, prediction accuracy improves with the increase in dataset size, suggesting that the model learns more effectively with more data. Notably, the algorithm demonstrates excellent scalability with large-scale datasets. Although training time and memory usage increase with the data size, the rate of increase remains relatively stable, indicating that the algorithm can maintain high efficiency and accuracy even under large-scale data processing. All curves have been smoothed to better illustrate the trends, with training time and memory usage presented on the left y-axis, and prediction accuracy displayed on the right y-axis.

As shown in Fig 15, with the increase in data volume, we observe a significant growth trend in processing time and response time. However, with the addition of computational resources (such as more CPU cores or GPUs), both processing time and response time are significantly optimized. This indicates that as computational resources scale, the algorithm can efficiently adapt to the processing demands of large-scale data. Notably, when using GPUs, the model's processing capability shows a marked improvement, further validating the algorithm's high scalability in large-scale data processing. By comparing the experimental results across different data volumes and computational resource configurations, it is evident that the algorithm can maintain high processing efficiency and response speed when dealing with big data, demonstrating its excellent scalability.

As shown in Figs 16 and 17, through a comparative experiment of accuracy and loss values before and after optimization, it is evident that the optimized algorithm demonstrates improvements in both accuracy and loss performance. Moreover, the optimized model exhibits more stable and superior performance across multiple training epochs. To achieve this optimization, we employed several strategies, including hyperparameter tuning (such as learning rate adjustment

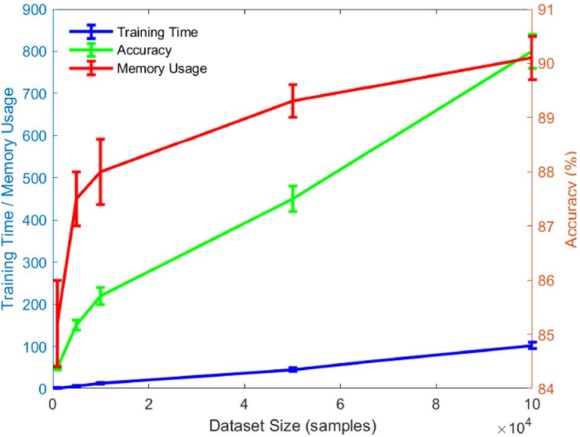

**Fig 14. Relationship Between Dataset Size, Training Time, Accuracy, and Memory Usage.**

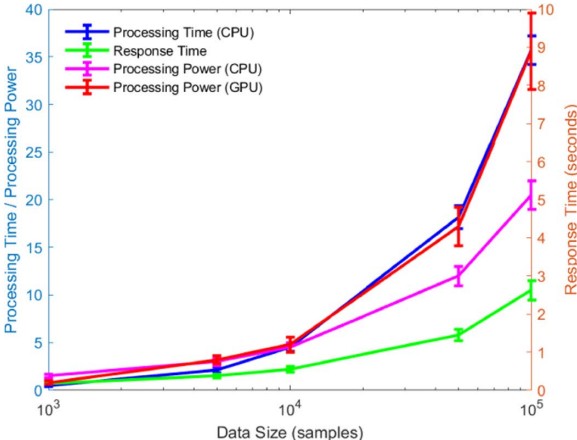

**Fig 15. Real-time Performance of Model with Different Data Sizes.**

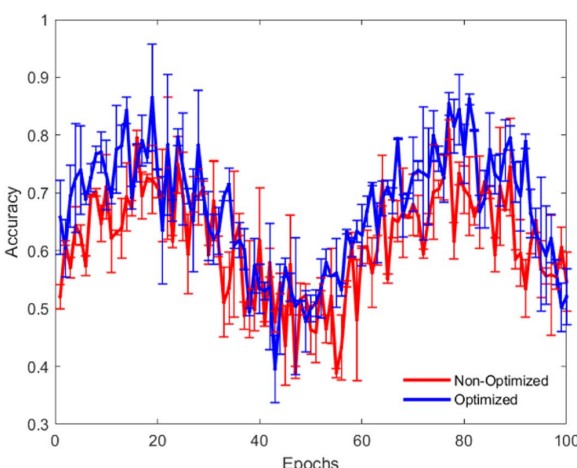

**Fig 16. Accuracy Comparison: Optimized vs Non-Optimized.**

and batch size modification), model architecture improvements (increasing network depth and width, introducing Batch Normalization and Dropout), efficient optimization algorithms (using the Adam optimizer with learning rate decay), data augmentation and regularization (enhancing data diversity and preventing overfitting), ensemble learning (combining multiple models to increase accuracy), and transfer learning (fine-tuning pre-trained models). These optimization methods significantly enhanced the model's accuracy and generalization ability, allowing the model to maintain high stability and superior performance across different data scales and training epochs, fully demonstrating the scalability of the algorithm, making it effective for large-scale applications. This indicates that our algorithm has strong scalability and can maintain high accuracy and low loss values, regardless of the dataset size or long-term training process.

As shown in Fig 18, the proposed deep learning-based model demonstrates strong scalability in different traffic environments. From the figure, it is clear that the model performs consistently across different cities and road types. Accuracy, recall, precision, and F1 score all maintain a high level. The error bars represent the performance fluctuation, which is small and stable across all environments. This indicates that the model's robustness is not significantly affected by

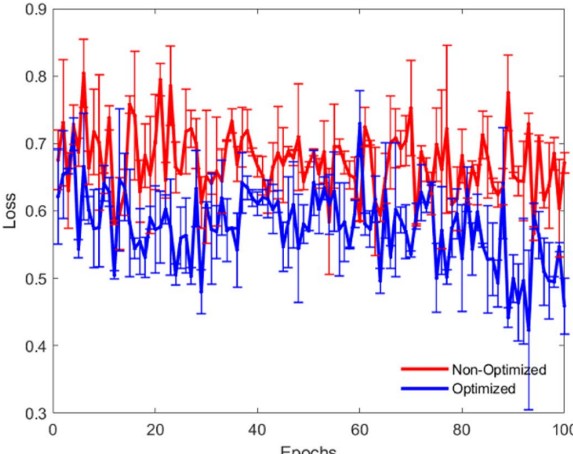

**Fig 17. Loss Comparison: Optimized vs Non-Optimized.**

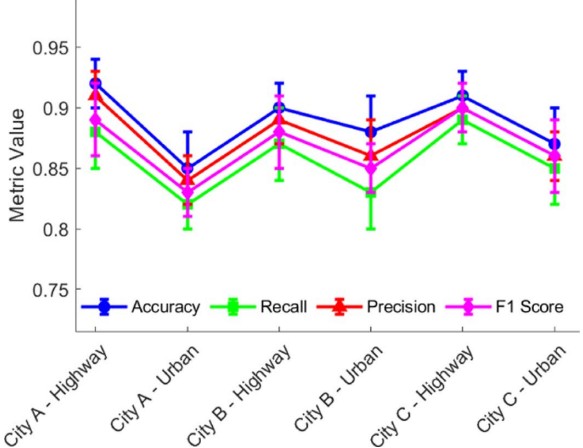

**Fig 18. Comparison of Model Parameters in Different Scenarios.**

differences in traffic conditions.This suggests that the algorithm can effectively generalize across various real-world scenarios, validating its application potential in different traffic environments. The consistent performance across different traffic environments further emphasizes the model's scalability, making it a strong candidate for wide application in traffic accident risk prediction.

As shown in Fig 19, with the extension of the time span, my combined algorithm demonstrates excellent and stable performance across various evaluation metrics, including accuracy, F1 score, precision, recall, and specificity. The error range is also relatively small.Specifically, as time progresses, the algorithm's metrics show a general improvement, and these changes exhibit a linear trend. This indicates that the algorithm can adapt to the demands of different time spans while maintaining high stability over extended periods.This stable performance trend suggests that the algorithm has strong scalability and can continue to perform efficiently in long-term predictions. Therefore, in practical applications, especially in scenarios involving long-term monitoring and forecasting, the algorithm shows strong reliability and adaptability.

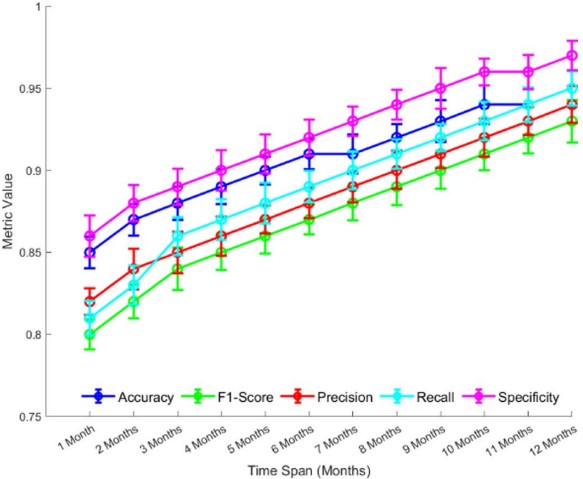

**Fig 19. Model Performance over 12 Months.**

The vehicle trajectory traffic accident risk prediction system based on deep learning and spatiotemporal features proposed in this study has significant scalability and can efficiently handle large-scale vehicle trajectory data. The system was designed using a distributed computing framework (such as Hadoop, Spark) and stream processing technologies (such as Apache Kafka, Flink) to ensure that massive time-series data can be processed efficiently with reduced latency, meeting various scaling needs.

As the data volume increases, the system can automatically scale computing resources to ensure high efficiency even under high concurrency. Additionally, the system architecture is highly flexible and can be adjusted according to different application scenarios, ensuring its adaptability in fields like intelligent transportation and autonomous driving.

In practice, we employed optimization strategies such as distributed training and transfer learning to improve the training efficiency and adaptability of the model on large-scale data. These methods not only reduce computing costs but also ensure that the model can be trained efficiently and perform real-time inference in the constantly changing traffic environment. Through these technologies, the system can quickly adapt to new data and tasks while ensuring high computational performance on various hardware platforms, further enhancing the system's scalability.

Given the uniqueness of traffic domain data, the system architecture not only supports efficient data processing but also combines edge computing and cloud platforms to enable real-time data processing and prediction. Edge computing reduces data transmission latency, ensuring that prediction results are generated within a short time, meeting the needs of real-time traffic management and autonomous driving systems. At the same time, the system can seamlessly integrate with existing intelligent traffic systems or autonomous driving platforms, further enhancing the system's practical value and flexibility.

To ensure stability and accuracy in different regions and environments, we designed a flexible optimization and retraining mechanism. By collecting multi-source data from different regions in real-time, the system can adaptively adjust model parameters to cope with varying traffic characteristics. Additionally, by regularly retraining the model, the system continuously optimizes and improves its performance on new tasks. These measures ensure that the system can handle changes in data and maintain stable prediction performance, further improving the scalability of the model.

However, the system faces challenges such as data privacy, security, and system reliability in practical applications. To address these challenges, we have implemented data encryption and anonymization technologies to ensure user privacy. Additionally, fault-tolerant mechanisms have been strengthened in the system design to ensure that the system

can quickly recover and maintain efficient operation in case of failure. We have also designed real-time monitoring tools to ensure the system's high availability and stability.

In conclusion, the traffic accident risk prediction system proposed in this study has powerful scalability and can meet the traffic data demands of different regions and scales. By closely collaborating with traffic management authorities and autonomous driving companies, we have further improved the system's accuracy and stability. The system not only has tremendous potential in intelligent traffic management and autonomous driving decision support but also can be quickly adjusted and optimized according to demand, promoting improvements in traffic safety and efficiency.

## Conclusion

This study proposes a traffic accident risk prediction method based on a CNN+LSTM+GNN combination model, which can automatically extract deep safety-related patterns from vehicle sensor data and traffic environment features to predict future collision risks in real-time. In real-world road scenario tests, the model demonstrates significant advantages over traditional machine learning methods, achieving accuracy and recall rates exceeding 90%, with specificity reaching 95.8%. Additionally, the model exhibits strong generalization ability across different road types, driving behaviors, and environmental conditions. Furthermore, experimental validation confirms the model's real-time performance in handling large-scale data, with an inference speed of 6000–7000 data points per second, meeting the requirements of real-time traffic monitoring.

Despite achieving significant results, this study has some limitations. First, the model training relies heavily on a large amount of labeled data, and the scarcity of traffic accident samples limits further improvement of the model. Second, the model lacks interpretability and has not yet provided clear causal explanations for accident risk predictions. Third, the model's online adaptability is insufficient, as it cannot be updated in real-time to address changes in the road network. Finally, although good inference efficiency has been achieved on edge devices, there is still room for optimization in terms of power consumption and response time.

To address the above limitations, future research will focus on incorporating active learning and semi-supervised learning techniques to improve learning efficiency for scarce accident samples and reduce labeling costs. At the same time, attention mechanisms and causal reasoning will be integrated to enhance the model's interpretability. Further development of incremental learning mechanisms will enable dynamic updates, improving the model's adaptability to real-time road conditions and environmental changes. The model structure will be optimized through lightweight design and efficient inference algorithms to enhance processing capability on edge devices, while incorporating external factors such as weather and traffic control to improve prediction accuracy. Additionally, collaborative sensing technologies will be explored to expand the model's applications in the Internet of Vehicles (IoV) and smart cities, enhancing traffic safety and early warning capabilities.

## Author contributions

**Conceptualization:** Hao Li.

**Data curation:** Linbing Chen.

**Formal analysis:** Linbing Chen.

**Funding acquisition:** Hao Li, Linbing Chen.

**Investigation:** Hao Li, Linbing Chen.

**Project administration:** Linbing Chen.

**Resources:** Linbing Chen.

**Writing – original draft:** Linbing Chen.

**Writing – review & editing:** Linbing Chen.

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
