## [Decision Letter · Decision Letter 0]

6 Jan 2025

PONE-D-24-57541Traffic Accident Risk Prediction based on Deep Learning and Spatiotemporal Features of Vehicle TrajectoriesPLOS ONE

Dear Dr. Chen,

Thank you for submitting your manuscript to PLOS ONE. After careful consideration, we feel that it has merit but does not fully meet PLOS ONE’s publication criteria as it currently stands. Therefore, we invite you to submit a revised version of the manuscript that addresses the points raised during the review process.

This study still needs significant revisions before qualifying for this journal publication.

We look forward to receiving your revised manuscript.

Kind regards,

Lei Zhang, PhD

Academic Editor

PLOS ONE

Journal Requirements:

Additional Editor Comments:

This study still needs significant revisions before qualifying for this journal publication.

Reviewers' comments:

Reviewer's Responses to Questions

**Comments to the Author**

1. Is the manuscript technically sound, and do the data support the conclusions?

Reviewer #1: Yes

Reviewer #2: Partly

2. Has the statistical analysis been performed appropriately and rigorously? 

Reviewer #1: Yes

Reviewer #2: Yes

3. Have the authors made all data underlying the findings in their manuscript fully available?

Reviewer #1: No

Reviewer #2: No

4. Is the manuscript presented in an intelligible fashion and written in standard English?

Reviewer #1: Yes

Reviewer #2: No

5. Review Comments to the Author

Reviewer #1: 1. All formulas in this article lack numbering to ensure that the main text and formulas comply with the journal's required format.

2. Lack of comparative experiments with state-of-the-art methods

3. Lack of visualization results for trajectory prediction and risk prediction

4. The innovative part of this article is not clear enough. Please present it in sections after the introduction

5. Please provide the parameters of the model designed in this article and the hyperparameters used during training

6. Insufficient summarization of existing research in the Introduction, it is recommended to cite " Research on multi-lane energy-saving driving strategy of connected electric vehicle based on vehicle speed prediction" and " Continual driver behavior learning for connected vehicles and intelligent transportation systems: Framework, survey and challenges"

Reviewer #2: I have the following comments and suggestions to further improve the quality of the work:

1) The abstract is not concise enough to sketch the entire theme, in particular, the results of the manuscript.

2) Furthermore, the introduction section needs considerable effort (concise and brief). The problem being investigated should be described clearly, but before that, the field of research should be made clearer. Furthermore, briefly describe the major contributions in bullet form, just before the organization paragraph.

The introduction should lead the way throughout the paper. In addition, the benefits coming from this paper should be made clearer in the introduction and throughout the paper.

3) I suggest summarizing the related works into a table with respect to their characteristics. The authors should put their proposal into this table for easy comparison. Furthermore, I have listed some related works that authors should consider for evaluation of their work.

This will make it clearer to readers, and they will be able to see what was missing in the literature and how this is addressed in this paper.

1. Exploiting Dynamic Spatio-temporal Graph Convolutional Neural Networks for Citywide Traffic Flows Prediction, Neural Networks

2. A data aggregation based approach to exploit dynamic spatio-temporal correlations for citywide crowd flows prediction in fog computing, Multimedia Tools and Applications

3. Exploiting dynamic spatio-temporal correlations for citywide traffic flow prediction using attention based neural networks, Information Sciences 577, 852-870

4. Modeling Dynamic Spatio-temporal Correlations for Urban Traffic Flows Prediction, IEEE Access

4) Do the results shown in various figures refer to a single run or multiple runs (average)? In the latter case, I will suggest adding standard deviation bars. The reason behind this is to ensure that the results overlap with the closest rivals or not.

5) Furthermore, the evaluation metrics should be briefly described in the experimental section. Moreover, add further details on how the experiments were conducted. Which tool was used to implement the algorithms? Similarly, system and resource characteristics should be added for clarity.

6) Some sections have repeated text or the discussion is too detailed and should be reduced. In fact, the paper is quite wordy, and the authors should make significant efforts to make it concise and short.

7) Add further details on how simulations were conducted. Perhaps add a flowchart that clearly identifies how the entire system works.

8) The conclusion section also needs significant revisions. It should briefly describe the findings of the study and some more directions for further research.

9) Proofread the article to ensure appropriate use of English grammar, tenses, and punctuation. Longer sentences should be broken out into smaller ones. There are also some linguistic issues that should be corrected. The use of article "the" is redundant and somewhere missing.

10) I suggest adding a brief description of each figure in its caption.

11) Authors should describe the scalability of the proposed algorithms. Is the system scalable? If yes, under what assumptions and conditions?

12) Add further details on how simulations were conducted. Perhaps add a flowchart that clearly identifies how the entire system works. The evaluation metrics should be briefly described in the experimental section. Moreover, add further details on how simulations were conducted. Similarly, system and resource characteristics could be added to tables for clarity.

13) More experiments should be performed to study the scalability of the proposed algorithms.

14) How can the proposed system be implemented in practice? I believe authors should consider this instead of giving unnecessary details of various systems and concepts.

15) The paper makes no mention of how the proposed algorithms and system will be implemented in practice, what the algorithm's computational cost is, or how it can be integrated into a real-world system.

16) Furthermore, what is being advanced in the state of the art and what originality is being introduced to the traffic prediction literature are both lacking in novelty. There is relatively little scientific contribution because this does not offer a new theory or model that is not already available.

17) Furthermore, each and every mathematical notation should be explained. I suggest adding a table that describes all the mathematical notations with brief definitions.

6. PLOS authors have the option to publish the peer review history of their article (what does this mean? ). If published, this will include your full peer review and any attached files.

**Do you want your identity to be public for this peer review?** For information about this choice, including consent withdrawal, please see our Privacy Policy .

Reviewer #1: No

Reviewer #2: No

---

## [Author Response · Author response to Decision Letter 1]

21 Jan 2025

Thank you for your valuable suggestions. I have made the revisions according to your requests.

---

## [Decision Letter · Decision Letter 1]

28 Jan 2025

PONE-D-24-57541R1Traffic Accident Risk Prediction based on Deep Learning and Spatiotemporal Features of Vehicle TrajectoriesPLOS ONE

Dear Dr. Chen,

Thank you for submitting your manuscript to PLOS ONE. After careful consideration, we feel that it has merit but does not fully meet PLOS ONE’s publication criteria as it currently stands. Therefore, we invite you to submit a revised version of the manuscript that addresses the points raised during the review process.

The revised manuscript has been largely improved compared to the original submission. But there are still some places that need further revisions before qualifying for this journal publication.

We look forward to receiving your revised manuscript.

Kind regards,

Lei Zhang, PhD

Academic Editor

PLOS ONE

Additional Editor Comments :

The revised manuscript has been largely improved compared to the original submission. But there are still some places that need further revisions before qualifying for this journal publication.

Reviewers' comments:

Reviewer's Responses to Questions

**Comments to the Author**

1. If the authors have adequately addressed your comments raised in a previous round of review and you feel that this manuscript is now acceptable for publication, you may indicate that here to bypass the “Comments to the Author” section, enter your conflict of interest statement in the “Confidential to Editor” section, and submit your "Accept" recommendation.

Reviewer #1: (No Response)

Reviewer #2: (No Response)

2. Is the manuscript technically sound, and do the data support the conclusions?

Reviewer #1: (No Response)

Reviewer #2: Yes

3. Has the statistical analysis been performed appropriately and rigorously? 

Reviewer #1: (No Response)

Reviewer #2: No

4. Have the authors made all data underlying the findings in their manuscript fully available?

Reviewer #1: (No Response)

Reviewer #2: Yes

5. Is the manuscript presented in an intelligible fashion and written in standard English?

Reviewer #1: (No Response)

Reviewer #2: Yes

6. Review Comments to the Author

Reviewer #1: (No Response)

Reviewer #2: There are duplicate references that should be removed, e.g. [13] and [14]

Exploiting Dynamic Spatio-temporal Graph Convolutional Neural Networks for Citywide Traffic Flows Prediction, Neural Networks

I suggest summarizing the related works into a table with respect to their characteristics. The authors should put their proposal into this table for easy comparison. Furthermore, I have listed some related works that authors should consider for evaluation of their work.

This will make it clearer to readers, and they will be able to see what was missing in the literature and how this is addressed in this paper.

Furthermore, every mathematical notation should be explained. I suggest adding a table that describes all the mathematical notations with brief definitions.

Proofread the article to ensure appropriate use of English grammar, tenses, and punctuation. Longer sentences should be broken out into smaller ones. There are also some linguistic issues that should be corrected. The use of article "the" is redundant and somewhere missing.

Furthermore, the evaluation metrics should be briefly described in the experimental section.

I believe the references and literature are not studied well and there are lots of important papers that address the same issues. I suggest authors include more related references and compare there results.

Li, Wei, et al. "Location and time embedded feature representation for spatiotemporal traffic prediction." Expert Systems with Applications 239 (2024): 122449.

Liu, Yutian, et al. "RT-GCN: Gaussian-based spatiotemporal graph convolutional network for robust traffic prediction." Information Fusion 102 (2024): 102078.

Implementation of augmented reality and drones to serve smart cities M AL-BAHRI, W Al Kishri, RR Dharamshi Artificial Intelligence & Robotics Development Journal, 147-157

Internet of Things: Layers, possible attacks, secure communications, challenges. A Alblushi, MJ Yousif Applied computing Journal, 103-118

ApMove: A Service Migration Technique for Connected and Autonomous Vehicles M Zakarya, L Gillam, AA Khan, O Rana, R Buyya IEEE Internet of Things Journal

Fan, Jin, et al. "RGDAN: A random graph diffusion attention network for traffic prediction." Neural networks 172 (2024): 106093.

Comparison of the proposed algorithm should be completed with other close rivals.

7. PLOS authors have the option to publish the peer review history of their article (what does this mean? ). If published, this will include your full peer review and any attached files.

**Do you want your identity to be public for this peer review?** For information about this choice, including consent withdrawal, please see our Privacy Policy .

Reviewer #1: No

Reviewer #2: No

---

## [Author Response · Author response to Decision Letter 2]

8 Feb 2025

This study has made modifications to the repeated citations and has followed your suggestions to categorize the relevant references. A table has been added to describe all mathematical symbols along with their brief definitions. The article's grammar has been revised, and a brief description of the experimental metrics has been included. The algorithms and models from the related literature you recommended have been studied and compared. Thank you very much for your guidance!

---

## [Decision Letter · Decision Letter 2]

13 Feb 2025

PONE-D-24-57541R2Traffic Accident Risk Prediction based on Deep Learning and Spatiotemporal Features of Vehicle TrajectoriesPLOS ONE

Dear Dr. Chen,

Thank you for submitting your manuscript to PLOS ONE. After careful consideration, we feel that it has merit but does not fully meet PLOS ONE’s publication criteria as it currently stands. Therefore, we invite you to submit a revised version of the manuscript that addresses the points raised during the review process.

We look forward to receiving your revised manuscript.

Kind regards,

Lei Zhang, PhD

Academic Editor

PLOS ONE

**Additional Editor Comments:**

Further revisions are still needed to enrich the literature review, present a comparison study and provide proper discussions on the results.

Reviewers' comments:

Reviewer's Responses to Questions

**Comments to the Author**

1. If the authors have adequately addressed your comments raised in a previous round of review and you feel that this manuscript is now acceptable for publication, you may indicate that here to bypass the “Comments to the Author” section, enter your conflict of interest statement in the “Confidential to Editor” section, and submit your "Accept" recommendation.

Reviewer #1: (No Response)

Reviewer #2: (No Response)

2. Is the manuscript technically sound, and do the data support the conclusions?

Reviewer #1: Yes

Reviewer #2: Partly

3. Has the statistical analysis been performed appropriately and rigorously? 

Reviewer #1: Yes

Reviewer #2: No

4. Have the authors made all data underlying the findings in their manuscript fully available?

Reviewer #1: (No Response)

Reviewer #2: Yes

5. Is the manuscript presented in an intelligible fashion and written in standard English?

Reviewer #1: (No Response)

Reviewer #2: No

6. Review Comments to the Author

**Reviewer #1: ** (No Response)

**Reviewer #2: ** The introduction section needs considerable effort (concise and brief). The problem being investigated should be described clearly, but before that, the field of research should be made clearer. Furthermore, briefly describe the major contributions in bullet form, just before the organization paragraph.

The introduction should lead the way throughout the paper. In addition, the benefits coming from this paper should be made clearer in the introduction and throughout the paper.

Proofread the article to ensure appropriate use of English grammar, tenses, and punctuation. Longer sentences should be broken out into smaller ones. There are also some linguistic issues that should be corrected. The use of the article "the" is redundant and somewhat missing. Moreover, longer sentences should be converted into short sentences to convey a clear message.

The table describing mathematical notation should be placed at an appropriate place—not at the beginning (before the abstract).

The related work should be placed in a separate section (not the introduction). More references must be added to improve the literature and also, the latest works must be chosen to do a comparison and evaluation of the proposed work.

A spatio-temporal deep learning approach to simulating conflict risk propagation on freeways with trajectory data

T Wang, YE Ge, Y Wang, W Chen - Accident Analysis & Prevention, 2024 - Elsevier

Liu, Yutian, et al. "RT-GCN: Gaussian-based spatiotemporal graph convolutional network for robust traffic prediction." Information Fusion 102 (2024): 102078.

Implementation of augmented reality and drones to serve smart cities M AL-BAHRI, W Al Kishri, RR Dharamshi Artificial Intelligence & Robotics Development Journal, 147-157

Internet of Things: Layers, possible attacks, secure communications, challenges. A Alblushi, MJ Yousif Applied computing Journal, 103-118

ApMove: A Service Migration Technique for Connected and Autonomous Vehicles M Zakarya, L Gillam, AA Khan, O Rana, R Buyya IEEE Internet of Things Journal

Fan, Jin, et al. "RGDAN: A random graph diffusion attention network for traffic prediction." Neural networks 172 (2024): 106093.

Many more.

Do the results shown in various figures refer to a single run or multiple runs (average)? In the latter case, I will suggest adding standard deviation bars. The reason behind this is to ensure that the results overlap with the closest rivals or not.

More experiments should be performed to study the scalability of the proposed algorithms.

How can the proposed system be implemented in practice? I believe authors should consider this instead of giving unnecessary details of various systems and concepts.

The paper makes no mention of how the proposed algorithms and system will be implemented in practice, what the algorithm's computational cost is, or how it can be integrated into a real-world system.

7. PLOS authors have the option to publish the peer review history of their article (what does this mean? ). If published, this will include your full peer review and any attached files.

**Do you want your identity to be public for this peer review?** For information about this choice, including consent withdrawal, please see our Privacy Policy .

Reviewer #1: No

Reviewer #2: **Yes: ** Muhammad Zakarya

---

## [Author Response · Author response to Decision Letter 3]

17 Feb 2025

This study has made modifications to the repeated citations and has followed your suggestions to categorize the relevant references. A table has been added to describe all mathematical symbols along with their brief definitions. The article's grammar has been revised, and a brief description of the experimental metrics has been included. The algorithms and models from the related literature you recommended have been studied and compared. Thank you very much for your guidance!

---

## [Decision Letter · Decision Letter 3]

24 Feb 2025

Traffic Accident Risk Prediction based on Deep Learning and Spatiotemporal Features of Vehicle Trajectories

PONE-D-24-57541R3

Dear Dr. Chen,

We’re pleased to inform you that your manuscript has been judged scientifically suitable for publication and will be formally accepted for publication once it meets all outstanding technical requirements.

Kind regards,

Lei Zhang, PhD

Academic Editor

PLOS ONE

Additional Editor Comments (optional):

The revised manuscript can be accepted for publication.

Reviewers' comments:

Reviewer's Responses to Questions

**Comments to the Author**

1. If the authors have adequately addressed your comments raised in a previous round of review and you feel that this manuscript is now acceptable for publication, you may indicate that here to bypass the “Comments to the Author” section, enter your conflict of interest statement in the “Confidential to Editor” section, and submit your "Accept" recommendation.

Reviewer #1: (No Response)

Reviewer #2: All comments have been addressed

2. Is the manuscript technically sound, and do the data support the conclusions?

Reviewer #1: (No Response)

Reviewer #2: Yes

3. Has the statistical analysis been performed appropriately and rigorously? 

Reviewer #1: (No Response)

Reviewer #2: Yes

4. Have the authors made all data underlying the findings in their manuscript fully available?

Reviewer #1: (No Response)

Reviewer #2: Yes

5. Is the manuscript presented in an intelligible fashion and written in standard English?

Reviewer #1: (No Response)

Reviewer #2: Yes

6. Review Comments to the Author

Reviewer #1: (No Response)

Reviewer #2: No more comments. Proofread the article to ensure appropriate use of English grammar, tenses, and punctuation. Longer sentences should be broken out into smaller ones.

7. PLOS authors have the option to publish the peer review history of their article (what does this mean? ). If published, this will include your full peer review and any attached files.

**Do you want your identity to be public for this peer review?** For information about this choice, including consent withdrawal, please see our Privacy Policy .

Reviewer #1: No

Reviewer #2: **Yes: ** Muhammad Zakarya

---

## [Editor Report · Acceptance letter]

PONE-D-24-57541R3

PLOS ONE

Dear Dr. Chen,

I'm pleased to inform you that your manuscript has been deemed suitable for publication in PLOS ONE. Congratulations! Your manuscript is now being handed over to our production team.

Kind regards,

on behalf of

Dr. Lei Zhang

Academic Editor

PLOS ONE